# Mitochondrial Dysfunction and Mitophagy in Fuchs Endothelial Corneal Dystrophy

**DOI:** 10.3390/cells10081888

**Published:** 2021-07-26

**Authors:** Varun Kumar, Ula V. Jurkunas

**Affiliations:** 1Schepens Eye Research Institute, Massachusetts Eye and Ear, Boston, MA 02114, USA; varun_kumar@meei.harvard.edu; 2Department of Ophthalmology, Harvard Medical School, Boston, MA 02115, USA

**Keywords:** CEnCs, mitochondrial dysfunction, DNA damage, mitophagy, FECD

## Abstract

Fuchs endothelial corneal dystrophy (FECD) is a genetically complex, heterogenous, age-related degenerative disease of corneal endothelial cells (CEnCs), occurring in the fifth decade of life with a higher incidence in females. It is characterized by extracellular matrix (ECM) protein deposition called corneal guttae, causing light glare and visual complaints in patients. Corneal transplantation is the only treatment option for FECD patients, which imposes a substantial socioeconomic burden. In FECD, CEnCs exhibit stress-induced senescence, oxidative stress, DNA damage, heightened reactive oxygen species (ROS) production, mitochondrial damage, and dysfunction as well as sustained endoplasmic reticulum (ER) stress. Among all of these, mitochondrial dysfunction involving altered mitochondrial bioenergetics and dynamics plays a critical role in FECD pathogenesis. Extreme stress initiates mitochondrial damage, leading to activation of autophagy, which involves clearance of damaged mitochondria called auto(mito)phagy. In this review, we discuss the role of mitochondrial dysfunction and mitophagy in FECD. This will provide insights into a novel mechanism of mitophagy in post-mitotic ocular cell loss and help us explore the potential treatment options for FECD.

## 1. Introduction

The corneal endothelial is the innermost layer of the cornea and plays an important role in maintaining water balance and clarity of the cornea. Fuchs endothelial corneal dystrophy (FECD) is the most common corneal endothelial dystrophy. It is a bilateral, genetically heterogeneous degenerative disease of CEnCs occurring in 4% of the U.S. population over 40 years of age, with a higher incidence in women [1,2,3,4]. It is characterized by the progressive decline of the CEnCs and the formation of extracellular matrix excrescences [5,6,7] in Descemet’s membrane (DM), called guttae, leading to corneal edema and loss of vision. Currently, the only treatment for FECD is corneal transplantation, which accounts for approximately 32,000 of the endothelial keratoplasties performed in the U.S. annually. It carries substantial economic and social burdens. Understanding the disease pathogenesis is essential for developing pharmacotherapeutic interventions to halt the disease. In FECD, CEnCs exhibits stress-induced senescence [8], oxidant-antioxidant imbalance [9], mitochondrial DNA damage [10] and dysfunction [11], sustained unfolded protein response (UPR) [12,13], and endoplasmic reticulum (ER) stress [14]. Among these factors, mitochondrial stress [15,16,17,18] plays an important role in FECD pathogenesis. Maintaining functional mitochondria is the key to the ion pump function in CEnCs. Excessive damage to the mitochondria leads to its selective degradation called auto(mito)phagy [15,18]. Herein, we will review the effects of mitochondrial dysfunction and auto(mito)phagy in CEnCs degeneration seen in FECD and will highlight the importance of mitochondria in developing novel therapeutics for the treatment of FECD.

## 2. Mitochondria in CEnCs

The corneal endothelium is one of the most metabolically active tissues in the body. It preserves relative dehydration of the cornea to maintain corneal clarity. CEnCs create a selective passive barrier through simple and facilitated diffusion by osmotic force. This allows the continuous flow of solutes and nutrients from aqueous humor to the cornea and vice versa, thereby maintaining adequate corneal hydration. To achieve this, CEnCs have primary ions pumps such as Na^+^K^+^ATPase, and many secondary ion transporters such as solute carrier family 4 member 11 (SLC4A11), sodium bicarbonate cotransporter-1, Na^+^: 2HCO3^−^ (pNBCe1), Na^+^:K^+^:2Cl cotransporter (NKCC1), Cl^−^/HCO3^−^ exchanger (AE2), sodium proton exchanger-1, and Na^+^/H^+^ (NHE1) [19,20]. The functioning of these primary ion pumps (Na^+^K^+^ATPase) requires a large amount of ATP generated by the mitochondria of CEnCs. Due to many ion pumps and very active endothelial cell metabolism, mitochondrial density is very high in CEnCs, second only to retinal photoreceptors [21]. Mitochondria, the powerhouse of the cells, regulate many physiological processes in CEnCs and play a pivotal role in their survival [22,23,24].

## 3. Mitochondrial DNA Damage in FECD

Mitochondrial DNA (mtDNA) damage occurs in many neurodegenerative disorders [25] such as Alzheimer’s, Parkinson’s, and Huntington’s diseases. It also occurs in retinal diseases [26] such as age-related macular degeneration, diabetic retinopathy, glaucoma, and in corneal diseases [27] such as keratoconus, Kearns Sayre Syndrome, and FECD [28,29]. In human FECD tissues, we found that 8-hydroxydeoxyguanosine (8-OHdG), a marker of oxidized DNA lesions, accumulated in mtDNA of CEnCs, suggesting increased oxidative mtDNA damage [9]. Further studies using long-amplicon-quantitative polymerase chain reaction (LA-qPCR) demonstrated that human FECD specimens had significantly decreased small mitochondrial copy number and increased DNA lesion frequency (indicative of damage) than the normal specimens [11]. Similarly, using real-time quantitative PCR (RT-PCR), Czarny et al. demonstrated a significant increase in mtDNA damage in the DM of FECD patients compared to the control [28]. Moreover, there were 4977 base pair (bp) common deletions in mtDNA for DM of FECD patients compared to the control [28]. However, in the same study, there was a significant increase in mtDNA copy number per cell in DM of human FECD tissues compared to the control. Thus, there was a discrepancy with respect to mtDNA copy number in this study compared with the previously described study above, which could be due to two different PCR techniques (i.e., LA-qPCR and RT-PCR). However, there was no significant difference in mtDNA copy number in peripheral blood lymphocytes (PBLs) [28]. Moreover, when exposed to hydrogen peroxide (H_2_O_2_) for the induction of oxidative stress, mtDNA copy number per cell decreased significantly in the PBLs of FECD patients compared to the control [28]. In FECD explants, Gendron et al. suggested increased mtDNA damage and telomere shortening compared to the control [29]. However, cultured CEnCs from both the healthy control and FECD patients restores mtDNA levels, telomere length, and oxidant-antioxidant gene balance and have equal sensitivity to oxidative stress-induced cell death [29].

As Fuchs is prevalent in females [2,3], Liu et al. analyzed mtDNA damage in the mouse model of ultraviolet A (UVA)-induced FECD for both sexes and found that mtDNA damage was significantly more in the mouse corneal endothelial cells (MCEnCs) at week 4 and 8 post-UVA in females compared to males, suggesting the female susceptibility to FECD [10]. However, mtDNA exhibited a similar extent of damage in both male and female mice at day 1 post-UVA, and it recovered at week 2 post-UVA [10]. For the in vitro studies, menadione (MN) induced oxidative stress to study mtDNA damage in normal (HCEnC-21T or HCECi: normal telomerase immortalized human corneal endothelial) and Fuchs (FECDi: immortalized FECD cell lines derived from FECD specimens) cell lines. Halilovic et al. demonstrated that the FECD cell line had significantly reduced the small mtDNA copy number compared to the normal control cell line and remained low in quantity after MN exposure [11]. MtDNA damage was significantly greater in the control cell lines than the untreated control and remained significantly high in the FECD cell line after MN exposure [11]. Miyajima et al. demonstrated a dysfunctional Nrf2-NQO1 axis, and specifically loss of NQO1 (NAD(P)H:quinone oxidoreductase 1) protein in FECD contributes to mitochondrial DNA damage and estrogen genotoxicity, explaining the higher incidence of FECD in females [30]. Treatment with MN and catechol estrogen (4-hydroxyestradiol;4-OHE_2_) in normal (HCEnC-21T) and Fuchs (FECD-SV-73F-74, FECD-SV-61F-18) cell lines generated estrogen-DNA depurinating adducts and mitochondrial DNA damage [30]. Moreover, NQO1−/− cells had increased estrogen-DNA adducts and mitochondrial DNA damage compared to NQO+/+ when treated with MN and 4-OHE_2_ [30]. NQO1 overexpression also reduced mtDNA damage in the HCEnC-21T cell line after treatment with MN and 4-OHE_2_ [30].

Epidemiological studies have suggested the role of different mtDNA variants in FECD. For example, a rare variant in the gene coding for mitochondria protein peripheral-type benzodiazepine receptor-associated protein 1 *(TSPOAP1)* is found in FECD patients without TCF4 repeat expansion [31]. However, 10398G allele and Haplogroup I confer protective effects for FECD. Specifically, patients with the mtDNA A3243G point mutation have CEnC polymegethism and mild guttae [17].

## 4. Mitochondrial Dysfunction in FECD

Mitochondrial stress and dysfunction [32,33,34] have been involved in many diseases such as neurodegenerative diseases [35], cancer [36], and diabetes [37]. It is central to aging processes [38] and critical for the degeneration of post-mitotic cells in other organs [39], similar to non-replicative CEnCs in FECD [40]. In general, mitochondrial dysfunction in FECD comprises aberrant mitochondrial bioenergetics and dynamics. Abnormal mitochondrial bioenergetics in FECD involves loss of mitochondrial membrane potential (MMP) [15], abnormal ATP production [11], and increased production of mitochondrial ROS [11]. Mitochondrial dynamics consists of a coordinated cycle of mitochondrial fusion and fission to maintain its shape and size. Abnormal mitochondrial dynamics in FECD comprises aberrant mitochondrial fusion including decline in mitofusin 2, loss of mitochondrial mass [15,16], and fission including increased mitochondrial fission proteins indicative of fragmentation [18], and activation of mitochondrial-mediated intrinsic apoptosis pathway [11].

Specifically, with regard to mitochondria bioenergetics, the central FECD human corneal tissue samples showed reduced cytochrome oxidase activity (a complex IV enzyme of the mitochondria electron transport chain), clinically associating with central corneal edema [41]. Similarly, FECD cell lines demonstrated the absence of electron transport subunits, complex I and V involved in mitochondrial bioenergetics [15]. Decreased/absent mitochondrial electron transport chain proteins in FECD may initiate the alteration in MMP. FECD cell lines showed increased susceptibility to MMP loss in response to the mitochondrial depolarization agent, m-chlorophenyl hydrazone (CCCP) [15]. Loss of MMP could lead to abnormal mitochondria bioenergetics, triggering abnormal ATP production, oxygen consumption, and increased mitochondria ROS in FECD. Specifically, Halilovic et al. demonstrated a significant decrease in MMP for the Fuchs cell line compared to normal control cell lines at the baseline. The Fuchs cell line demonstrated a further decrease in MMP after treatment with MN compared to normal control cell lines [11]. Moreover, Halilovic et al. also showed that ATP production was significantly lower in the FECD cell line compared to control cell lines at the baseline [11]. The Fuchs cell lines exhibited greater susceptibility to MN-induced ATP loss compared to the control cell lines. The FECD cell line had significantly higher mitochondrial ROS at the baseline compared to control cell lines [11]. MN induced a further significant increase in mitochondria ROS for FECD cell line [11] compared to thee control. A general potent antioxidant, N-acetylcysteine (NAC), significantly rescued mitochondrial ROS in the control cell lines after MN treatment but could not rescue in FECD cell lines, indicating increased baseline oxidant-antioxidant imbalance for FECD [11].

Concerning aberrant mitochondria dynamics in FECD, Halilovic et al. also demonstrated mitochondrial fragmentation demonstrated by cytochrome c release in human Fuchs specimen compared to the normal control [11]. MN significantly increased cytochrome c release with time in the normal control cell line compared to untreated cells [11]. Mitochondrial fragmentation might lead to loss of mitochondrial mass in FECD. Benischke et al. showed the loss of mitochondrial mass in the Fuchs cell line compared to the control cell line at the baseline [15]. Mitochondrial depolarizing agent, CCCP, significantly decreased mitochondrial mass in the control cell line [15]. Altered mitochondrial dynamics such as mitochondrial mass loss and fragmentation may lead to the activation of the mitochondrial-mediated intrinsic pathway in FECD. The intrinsic apoptotic pathway is activated by intracellular stressors such as DNA damage and oxidative stress. This results in MMP loss and cytochrome c release, and subsequent apoptosome formation with caspase 9 activation. Finally, it leads to further initiation of executioner caspase 3/6/7 activation, ultimately resulting in apoptosis. Intrinsic apoptotic markers, caspase 9 and caspase 3, were significantly upregulated for the Fuchs cell line compared to the control cell line at baseline [11]. Moreover, normal control cell lines exhibited upregulation of caspase 9 and 3 upon treatment with MN [11], suggesting activation of the mitochondrial-mediated intrinsic apoptotic pathway after oxidative stress.

Despite several studies on mitochondrial DNA damage and dysfunctional mitochondria in FECD, there has not been any study directly associating the two phenomena in FECD. However, in other ocular [26] and neurological diseases [25], mitochondrial DNA damage is associated with mitochondrial dysfunction and disease pathogenesis. Based on mitochondrial dysfunction in many ocular and neurological diseases, mtDNA damage can be one of the major contributing factors for mitochondrial dysfunction in FECD.

## 5. Autophagy and Mitophagy

Cellular homeostasis requires balancing two important pathways: (a) catabolism, the breakdown of complex molecules inside a living organism’s cell and releasing energy, and (b) anabolism, the synthesis of complex molecules including the storage of energy. An imbalance of these two pathways produces large amounts of unwanted molecules and proteins in organelles in a cell, thereby activating cellular degradation pathways under pathological conditions. For large-scale cellular degradation, autophagy is one of the dominant intracellular degradative processes that exports cytoplasmic proteins to lysosomes for degradation. Autophagy is an evolutionarily conserved, genetically controlled natural mechanism of the cell to remove the cell’s unnecessary dysfunctional components/organelles, allowing constant recycling. Under normal physiological conditions, autophagy eliminates unwanted proteins or damaged organelles for maintaining cellular homeostasis. Under acute stress, autophagy can act as an adaptive protective mechanism by removing detrimental organelles or proteins involved in many diseases [42]. Mitophagy is a specialized and evolutionarily conserved autophagy involving degradation of damaged mitochondria, which accumulates following stress, thus regulating mitochondrial turnover and quality control and health. Mitochondria quality control is a complex cellular protective mechanism and occurs through the coordination of many cellular processes such as proteostasis, biogenesis, and dynamics including mitophagy to maintain cell homeostasis. Imbalance of mitochondrial quality control occurs in many diseases in response to mtDNA damage upon various intracellular or extracellular stress. Under extreme stress, when there is minimum possibility of maintaining cell homeostasis through mitochondria quality control, cells activate their apoptotic pathways to control cell fate.

## 6. Mechanisms of Mitophagy in FECD

Few studies have demonstrated the role of mitophagy in the pathogenesis of FECD [15,43] (Figure 1). Specifically, Benischke et al. demonstrated abnormal, swollen mitochondria in vacuoles called autophagosomes, suggesting initial signs of mitophagy in FECD human tissues using transmission electron microscopy (TEM) [15]. Initial features of mitophagy have also been reported in the UVA-induced mouse model of FECD [10] and Col8a2 (a mouse model of early-onset FECD) [43]. In the UVA-model [10], large autophagic vacuolar structures contained degraded mitochondrial in male and female MCEnCs at three months post-UVA [10]. In the Col8a2Q^(455K/Q455K)^ mouse model of early-onset FECD [43], 20-week old Col8a2Q^(Q455K/Q455K)^ mouse exhibited abnormal accumulation of mitochondria in the corneal endothelium, which could be initiating mitophagy. In the same study, a 20-week-old Col8a2Q^(L450W/L455W)^ mouse showed large vacuolar structures containing degraded mitochondria [43]. The mechanisms of mitophagy in the mouse model of early-onset FECD as well as the UVA-induced mouse model of FECD are still unclear. However, there have been few studies investigating the mechanism of mitophagy in the UVA-induced FECD [15,18] in vitro.

In general, mitophagy could be driven by the loss of MMP, electron transport chain proteins, and mitochondrial dysfunction, as seen in FECD as described in the previous section. This could lead to a significant decline in mitochondrial mass as seen in the FECD cell line compared with the control cell line (HCECi) [15] at the baseline. Moreover, CCCP could lead to a significant decline in mitochondrial mass in the control cell line (HCECi) [15]. Loss of mitochondria mass could directly contribute to mitophagy in FECD. Specifically, studies using a specific autophagy inhibitor, bafilomycin (a chemical that inhibits autophagosomes with lysosomes, i.e., autophagy), showed that mitochondrial mass could be rescued with bafilomycin after CCCP treatment in the control and FECD cell line [15]. One of the earliest markers for the occurrence of autophagy is the activation of autophagy markers such as a microtubule associated protein 1 light chain 3 (LC3), autolysosome-associated membrane protein 1(LAMP1) in FECD. LC3-II is a critical component of autophagosomes that gets integrated into the membrane following conjugation of the cytosolic LC3-I with phosphatidylethanolamine. Autophagy marker LC-II to -I ratio and autolysosome-associated membrane protein 1(LAMP1) was significantly increased in FECD specimens compared to the normal specimen [15]. Similarly, the FECD cell line showed a significant increase in LC3-II compared to the control cell line. The FECD cell line exhibited a further increase in LC3-II after treatment with CCCP, specifically in the mitochondria, possibly suggesting early signs of autophagy of mitochondria (i.e., mitophagy) [15]. Furthermore, mitophagy drives the loss of mitochondrial fusion protein such as Mitofusin 2 (Mfn2), known for stabilizing interaction between mitochondria in FECD. Benischke et al. demonstrated a significant reduction in Mfn2 protein for human FECD specimens. The FECD cell lines had a significant decrease in Mfn2 compared to the control cell line at the baseline, which further declined specifically in the mitochondria after treatment with CCCP. Using immunocytochemistry study for LC3 with Mfn2, Mfn2 co-localizes more with LC3 after treatment of CCCP and autophagy inhibitor (bafilomycin), suggesting that the activated autophagy degradation pathway probably led to Mfn2 protein degradation in FECD. In conclusion, autophagy in mitochondria or mitophagy drives the loss of Mfn2 in FECD [15].

Another molecular mechanism for mitophagy in FECD is PINK1(PTEN-induced putative kinase 1)-Parkin pathway activation, as described by our group [18]. In the PINK1-Parkin mediated mitophagy, there was upregulation of PINK1 in the outer mitochondrial membrane, which recruits the cytosolic E3 ubiquitin ligase Parkin and ubiquitin by phosphorylating Serine65 in both proteins. Parkin translocate to the mitochondria, and further ubiquitinates its substrates to proteasomal degradation. The ubiquitination also initiates the recruitment of phagosomes to mitochondria, removing them through the autophagosomal pathway. Miyai et al. showed that human FECD samples demonstrated upregulation of PINK1, phosphor-Parkin with a decrease in total Parkin [18]. A similar observation was seen at least for phosphor-Parkin and Parkin in normal cell lines after the MN induced mitochondrial fragmentation in vitro [18]. FECD cell lines also demonstrated a significant decrease in the total Parkin with significant upregulation of the phosphor-Parkin (Ser65) compared to the control cell line when treated with CCCP. To investigate the ubiquitination and proteasomal degradation of mitochondrial following Parkin recruitment, a proteasomal inhibitor (Epoxomicin) was used and demonstrated increased fold in the rescue of total Parkin after treatment of CCCP in the Fuchs cell line compared to the untreated control cell line. This suggests a possible abnormal stimulation of proteasome-mediated degradation in FECD. To further investigate, autophagy inhibitor (bafilomycin) rescued MN-dependent decrease in endogenous PINK1 and total Parkin, followed by a reversal of the increase in phosphor-Parkin (Ser65). These results indicate the reversal of Parkin activation by bafilomycin and suggest mitophagy-dependent degradation of endogenous proteins under oxidative stress [9].

Apart from mitophagy, autophagy is also reported in a mouse model of early-onset FECD. Kim et al. demonstrated that lithium treatment increases survival of CEnCs, possibly by upregulating autophagy [44]. Specifically, there were significantly more MCEnCs with regular cell size and less corneal guttae in the Col8a2Q^(455K/Q455K)^ mouse after treatment with lithium compared with the untreated control. Moreover, there was upregulation of autophagy-related genes (*P62, Tmem74, Tm9sf1, Tmem166*) and autophagy markers (Atg5-12) in the lithium-treated Col8a2Q^(455K/Q455K)^ mouse compared with the untreated control, suggesting its role in CEnCs survival.

## 7. Role of Mitophagy in FECD and Other Neurodegenerative Diseases

The activation of mitophagy being detrimental or useful remains unclear in FECD. PINK1-Parkin mediated mitophagy, as described by our group, suggests that excessive mitophagy might destroy many normal mitochondrial, disturbing mitochondrial bioenergetics, and contributing to the cascade of destructive events in the pathogenesis of FECD. Therefore, therapeutics targeting mitophagy might not be beneficial in the late stage of FECD. However, it might be advantageous in the earlier stages of FECD as CEnCs might alleviate some intracellular mitochondrial stress by removing damaged mitochondria via mitophagy.

CEnCs embryonically derived from the neural crest are non-replicative cells, like neurons, and degenerate with age in FECD, similar to the loss of retinal ganglion cells or neurons in many ocular or neurodegenerative diseases. Thus, understanding the mitophagy mechanism in neurodegenerative diseases can be beneficial and applicable in FECD. Defective mitophagy, leading to the accumulation of abnormal proteins and damaged mitochondria, occurs in many neurodegenerative diseases including Parkinson’s [45], Alzheimer’s [46], Huntington’s [47], and amyotrophic lateral sclerosis (ALS) [48]. Thus, activation and maintenance of mitophagy is beneficial in these neurodegenerative diseases and provides treatment options for neurological diseases. Specifically, in Alzheimer’s disease, Fang et al. demonstrated that by stimulating mitophagy by Nicotinamide adenine dinucleotide (NAD+) supplementation, urolithin A and actinonin reverse memory impairment by reducing extracellular plaques and neuroinflammation in the mouse model of Alzheimer’s [49]. In the drosophila model of Huntington’s disease, Khalil et al. demonstrated that mitophagy upregulation by the PINK-Parkin molecular pathway improves mitochondrial integrity and neuroprotection [50]. Similarly, in Parkinson’s disease, mitophagy upregulation by Mitochondria receptor Nip3-like protein X (Nix) restores mitochondrial function to protect against PINK1/Parkin related Parkinson’s disease [51]. Moreover, Hwang et al. exhibited that Glyceraldehyde-3-Phosphate Dehydrogenase (GAPDH)-induced mitophagy remains beneficial, however, its impairment contributes to Huntington’s disease [47].

However, excessive mitophagy might also be detrimental for the neurons in neurological diseases [52]. In Huntington’s disease, striatal neurons are preferentially vulnerable to 3-nitropropionic acid (3-NP), a mitochondrial complex-II inhibitor. A protein enriched in the striatum called Rhes removes damaged mitochondria via mitophagy. The mitophagy intensifies in the presence of 3-NP, suggesting exaggerated mitophagy to be a contributing factor to Huntington’s pathogenesis [52]. Similarly, Su et al. showed that excessive mitophagy and autophagy contributes to chronic cerebral hypoperfusion-induced neuronal death, and its inhibition is valuable in preventing the disease [53]. All these mitophagy studies suggest that neurons need to clear the damaged proteins and their associated organelles like mitochondria to enhance its survival; however, excessive clearance of mitochondria via mitophagy could be detrimental. Therefore, mitophagy behaves like a double-edged sword. Thus, simple upregulation of mitophagy might not be a feasible approach. Mitophagy activities must be optimized to balance neuroprotection in neurological diseases, which might be applied in FECD. The detailed mechanisms of mitophagy need to be strengthened. The interactions and molecular mechanisms involving mitochondrial bioenergetics, dynamics, and mitophagy require detailed investigation.

## 8. Unanswered Questions

(a)Is the activation of mitophagy beneficial or detrimental to CEnC survival, thereby playing a bidirectional role in FECD?(b)How does lysosome play a role in the clearance of autophagic structures containing mitochondria in FECD?(c)How are the mechanisms of mitophagy in FECD different from neurodegenerative diseases or cancer?(d)Does mitophagy play a context-dependent role at various stages of FECD?(e)How can we use mitophagy as a therapeutic option for FECD?

## 9. Conclusions

In this review, we have provided an overview of mitochondrial dysfunction and its close relationship with mitophagy in FECD. Altered mitochondrial bioenergetics such as decreased ATP production, loss of MMP, increased mitochondria ROS production, and abnormal mitochondrial dynamics such as loss of mitochondrial mass, increased fragmentation, and DNA damage is involved in the pathogenesis of FECD. Moreover, excessive mitochondrial damage in FECD could form autophagic vacuoles containing swollen degenerating mitochondria, which are removed by mitophagy. Specifically, mitophagy results in the loss of mitochondrial fusion protein such as Mfn2 as well as mitochondrial mass in FECD. There is an activation of PINK1-Parkin-mediated mitophagy at the molecular level, which degrades mitochondria control proteins in FECD. However, it is still unclear whether mitophagy is beneficial or detrimental to CEnC survival in FECD. Future studies are needed to investigate the role of accumulating autophagic vacuoles containing mitochondria in FECD pathogenesis.

## Figures and Tables

**Figure 1 cells-10-01888-f001:**
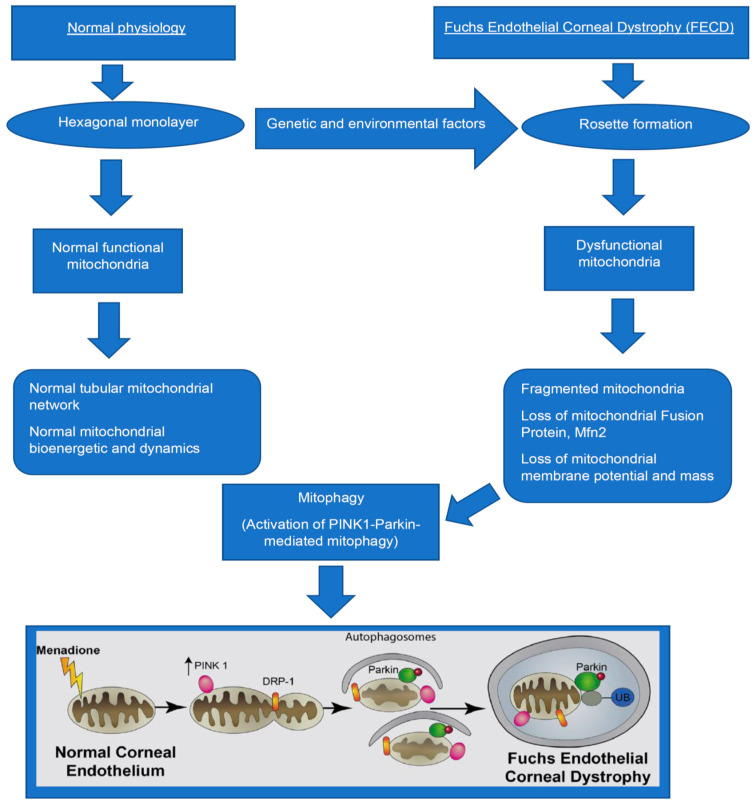
Mitophagy in Fuchs endothelial corneal dystrophy. Schematic diagram demonstrating the formation of dysfunctional mitochondria that leads to mitophagy in Fuchs. In FECD, various genetic and environmental factors lead to the conversion of hexagonal CEnCs into an irregular shape, thereby resulting in rosette formation. Subsequently, there is mitochondria DNA damage, fragmentation, loss of mitochondria fusion protein (Mfn2), mass and membrane potential, which lead to activation of PINK1-Parkin mediated mitophagy, thereby resulting in clearance of abnormal mitochondria.

## Data Availability

Not applicable.

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
