# Peer review of "Mitochondrial Dysfunction and Mitophagy in Fuchs Endothelial Corneal Dystrophy"

_cells, 2021, doi:10.3390/cells10081888_

Round 1

Reviewer 1 Report

The review is very well written and easy to read. It provides an excellent overview of information on the subject of Fuchs corneal endothelial dystrophy in relation to mitochondrial dysfunction and mitophagy. It nicely puts the previous results on the subject in perspective. I particularly appreciated the care taken by the authors to put them into perspective with respect to the models used in the different articles, avoiding overstatement in a nice way.

My main criticism is on the level of novelty. I find it difficult to determine what this review brings more than the reviews published by the same corresponding author in recent years. In particular, the 2 reviews published in 2021 in Prog Retin Eye Res.

Recently, an article was published by the authors of this review (FRBM 2021). The latter is directly related to the review presented and it seemed surprising to me that it was not discussed.

Finally, an article by the author is cited extensively in the review (i.e. Benischke et al Sci Rep 2017), as if much of the review serves to put this article in a global context of the pathology.

Author Response

Please see the attachment below.

Reviewer 2 Report

The manuscript by Kumar and Jurkunas is a review discussing the role of mitochondrial dysfunction and mitophagy as emerging targets in Fuchs endothelial corneal dystrophy. The manuscript is fairly well written in general and is a nice “wrap-up” of work in the field, in which the authors are knowledgeable. It does focus a lot on the authors previous works, which is however understandable given that the literature regarding this particular topic is also not very broad.

Nevertheless, there are some points that the authors should address in order to improve the manuscript:

The authors should refrain from using indiscriminately the term auto(mito)phagy. Please use either autophagy or mitophagy depending on the specific process they refer to in a given sentence.

The concept of mitochondrial dynamics should be made clear in a sentence to be added to the text. Right now the authors refer to “mitochondrial fragmentation demonstrated by cyt c release” (line 139) and “altered mitochondrial dynamics such as mitochondrial mass loss and fragmentation” (line 146) and the concepts of fission and fusion, for instance, are not clearly defined, and are only referred later (line 179). And even mitophagy, which is central in the manuscript, is only defined later in the text (line175) and it is defined as “the selective degradation of mitochondria by autophagy, thereby creating an imbalance of mitochondrial quality control.” Moreover, I disagree with the statement “creating an imbalance of mitochondrial quality control” as mitophagy occurs precisely to restore a certain balance do the mitochondrial network. Additionally the authors state in a following sentence that “Altered mitochondrial quality control could lead to mitophagy”…I advise a rephrasing of this paragraph for more consistency.

On line 213 “mitophagy inhibitor, bafilomycin (a chemical that inhibits autophagosomes with lysosomes i.e., mitophagy)”. Not only is missing words “that inhibits the fusion of autophagosomes with lysosomes” as it is incorrectly stated as “ i.e. mitophagy”(mitophagy is not the fusion of autophagosomes with lysosomes). And on line 251 bafilomycin is stated as a mitophagy inhibitor which is not accurate. Bafilomycin inhibits autophagy in general, and not only mitophagy.

On line 216 “One of the earliest mechanisms of mitophagy is the activation of autophagy markers such as a microtubule associated protein 1 light chain 3 (LC3), autolysosome-associated membrane..” Sentence should be rephrased: activation of autophagy markers is not a mechanism of mitophagy. At most activation of autophagy markers can be one of the earliest signs/markers of the occurrence of mitophagy, and still there are other more specific markers of mitophagy, as LC3 and LAMP1 are markers of autophagy, and not specifically of mitophagy.

Author Response

Please see the attachment below.

Round 2

Reviewer 1 Report

I am still not convinced by the novelty of this review. But the authors answered appropriately to the main concerns I had.